# Recording of patients' mental health and quality of life-related outcomes in primary care: a cross-sectional study in the UK

Helena Carreira ,[1] Rachael Williams,[2] Harley Dempsey,[2] Krishnan Bhaskaran[1]

¹Department of Non-Communicable Disease Epidemiology, London School of Hygiene & Tropical Medicine, London, UK
²Clinical Practice Research Datalink, Medicines and Healthcare Products Regulatory Agency, London, UK

**Correspondence to**
Helena Carreira;
helena.carreira@lshtm.ac.uk

## ABSTRACT

**Objective** To compare patient-reported anxiety, depression and quality-of-life (QoL) outcomes, with data registered in patients' primary care electronic health record (EHR).

**Design** Cross-sectional study.

**Setting** Primary care in the UK.

**Participants** A convenience sample of 608 women registered in the Clinical Practice Research Datalink GOLD primary care database (data from a previous study on 356 breast cancer survivors (8.1 years postdiagnosis) and 252 women with no prior cancer).

**Outcome measures** Patient-reported data on anxiety, depression and QoL, collected through postal questionnaires, and compared with coded information in EHR up to 2 years prior.

**Results** Abnormal anxiety symptoms were reported by 118 of 599 women who answered the relevant questions (21%); 59/118 (50%) had general practitioner (GP)-recorded anxiolytic/antidepressant use, and 2 (1.6%) had anxiety coded in the EHR. 26/601 women (11%) reported depression symptoms, of whom 17 (65.4%) had GP-recorded antidepressant use and none had depression coded. 65 of 123 women reporting distress on the pain QoL domain (52.8%) had a corresponding record in the EHR <3 months before and 92 (74.8%) <24 months before. No patients reporting fatigue (n=157), sexual health problems (156), social avoidance (82) or cognitive problems (93) had corresponding codes in the EHR. There were no meaningful differences in the concordance results between breast cancer survivors and women with no history of cancer.

**Conclusion** Many patients reporting mental health and QoL problems had no record of this in coded primary care data. This finding suggests that coded data does not fully reflect the burden of disease. Further research is needed to understand whether or not GPs are aware of patient distress in cases where codes have not been recorded.

### STRENGTHS AND LIMITATIONS OF THIS STUDY

⇒ A strength of this study comes from the use of the Clinical Practice Research Datalink (CPRD) GOLD primary care database to select participants for the study, as it enabled the comparison of patient-reported outcomes (PROs) with the data that had been routinely recorded in their electronic health record (EHR).

⇒ PROs were assessed using validated tools and identification of data in the coded EHRs was based on a systematic review of Read codes.

⇒ Limitations of this study include the lack of information for drugs sold over the counter, which are not captured by CPRD, and that we could not distinguish when anxiolytics and antidepressants may have been used for conditions other than anxiety/depression.

⇒ Most patient care is recorded using codes but general practitioners sometimes use other methods to keep records (eg, free text entry) which are not available to us; similarly, some codes in the patient records are unspecific (eg, "mood observations") and we could not assign a correspondence to domains of quality-of-life.

⇒ This study included only a convenience sample of adult women and the results may not be generalisable to all women, men or to other age groups.

should include prevention, timely diagnosis, and optimising management and treatment of prevalent cases.[3]

Early diagnosis and treatment of patients with adverse mental health outcomes are not always possible, in part because symptoms are often unspecific and go unrecognised, and because patients do not always seek care for mental health-related conditions.[4 5] There has been a lack of research quantifying the burden of mental health and other QoL-related complaints that have not been picked up in primary care, and therefore may remain undiagnosed and untreated.[6] One way of quantifying the gap between adverse mental health and QoL-related outcomes recorded

## INTRODUCTION

Quality of life (QoL) and mental health are among the most important outcomes for individuals, but the prevalence of problems is high.[1 2] Improving QoL and reducing the mental health burden is challenging but there is consensus that public health strategies

in primary care, and those experienced by patients, is to directly collect patient-reported outcomes (PROs) and compare with information on the same outcomes in the clinical record. Under-recording of problems in primary care records could suggest lack of awareness by the general practitioner (GP) about the patient's lack of well-being, and thus a missed opportunity for care. Under-recording might also reflect inconsistent coding of mental health and QoL problems in the primary care record, with important implications for audit and research based on electronic health records (EHR).[7 8]

In this study, we compared patient-reported information on symptoms of anxiety, depression and QoL domains, with data for similar constructs registered in the patients' EHR. We used a convenience sample of data from a previous study that collected PROs data from women with and without a history of breast cancer,[9] and for whom EHR data were also available.

## METHODS
### Study design and sampling frame
We used a convenience sample of women with PRO data available from a previous study.[9] For the original study, primary care practices contributing with data to the Clinical Practice Research Datalink (CPRD) GOLD primary care database in August 2018 were invited to participate. CPRD GOLD includes EHR of patients attending general practices in the UK that use Vision software to manage patient's records. Data are entered in the patients' EHR by GPs during consultations using Read codes,[10] which include information on symptoms, diagnoses and prescriptions.[11] The study protocol (online supplemental materials) provides the sample size calculations for the original study. Patients registered with primary care practices that accepted to participate were considered potentially eligible for the study.

### Patient eligibility criteria, selection and recruitment
A full description of eligibility and recruitment has been published elsewhere.[9] Briefly, inclusion criteria for the breast cancer survivors' group were (1) diagnosis of invasive breast cancer at least 1 year before (all stages) and (2) aged 18–80 years. To ensure that the recorded breast cancer was incident, we required 1 year of follow-up in CPRD prior to the breast cancer diagnosis. For the comparison group, inclusion criteria were (1) no history of cancer (except non-melanoma skin cancer), (2) aged 18–80 years and (3) at least 2 years of follow-up data in CPRD (since we required 1 year of follow-up before and after cancer to be included in the breast cancer group). Exclusion criteria for both groups were (1) inability to complete a self-reported questionnaire (eg, due to dementia) and (2) having had another (non-breast) cancer or having been treated for a non-invasive breast tumour.

The CPRD GOLD primary care database was used to identify all breast cancer survivors from participating practices, and a random sample of women with no prior cancer (frequency matched on age to breast cancer survivors) from the same practices. GPs applied inclusion and exclusion criteria (vide above), and sent the study materials to the eligible patients' addresses with a prepaid envelope to return the questionnaires. Patients were recruited between January and November 2019.

### Patient-reported outcomes
#### Anxiety and depression
Anxiety and depressive symptoms were measured with the Hospital Anxiety and Depression Scale.[12] This is a 14-item self-reported screening tool for anxiety and depressive symptoms in the past week.[12] Based on their responses, we categorised patients as non-case (scores 0–7), borderline (scores 8–10) and probable case (scores 11–21).[12]

The QoL impact of anxiety and depression was measured with the respective domains in the Quality of Life in Adult Cancer Survivors Scale (QLACS) (see below).

#### Quality of life
QoL was assessed with QLACS.[13] This tool includes 47 items, divided into 7 generic domains (ie, negative feelings; positive feelings; cognitive problems; pain; sexual function/interest; energy/fatigue and social avoidance), and 5 cancer-specific domains (ie, financial problems; benefits of cancer; distress-family; appearance; distress-recurrence) which are not considered further in this paper. Of the seven generic QoL domains, six were considered suitable for comparison with data in the EHR because women with distress for these domains may visit their GP to seek help: 'negative feelings', 'cognitive problems', 'pain', 'sexual problems', 'fatigue' and 'social avoidance'. Each domain considered has four items on the QLACS questionnaire. Participants are instructed to answer in relation to the previous 4 weeks. Responses to each item are given on a Likert-type of scale that varies between 1 (never) and 7 (always); higher scores indicate poorer QoL.

To identify women who had high levels of distress for each domain, we calculated the mean response (ie, the sum of the individual item scores divided by four; mean values range between one and seven). We considered a mean of ≥5 (corresponding to average replies of frequently, very often or always experiencing the stated symptom) to reflect distress in that domain. This was varied in sensitivity analyses (see below).

### Outcomes recorded in EHRs primary care data
We extracted the primary care EHR data for all participants. As PROs were collected between January and November 2019, we extracted data from the January 2020 version of CPRD GOLD, which included data from 1987 up to December 2019.

Anxiety and depression were defined using lists of Read codes from a systematic review.[14] For the QoL domains, we produced lists of Read codes closely related to the items in the QLACS domain (online supplemental table

1). Read codes were used to identify women with these codes registered in their EHR in the 3, 6, 12 and 24 months prior to the date of last data collection from the practice. The last collection date varied from practice to practice, but was generally within 3 weeks of the database version (eg, in the January 2020 version, the date of last data collection from the practices was in median of 20 days (IQR: 19–20) prior to 31 December 2019).

## Data analysis

We calculated the proportion of women who reported high levels of distress in the questionnaires and had similar information in their EHR (ie, sensitivity of the EHR for capturing patient-reported distress). To better understand the agreement between PROs and the EHR data, as a secondary analysis, we also calculated the proportion of women with codes indicating distress in each domain in their EHR that reported distress levels in the questionnaires (positive predictive value of the EHR for capturing patient-reported distress). Results were shown in tables and descriptively.

## Sensitivity analysis

As we used an arbitrary cut-off to identify patients with poor QoL (mean domain-specific score of ≥5), two sensitivity analyses were conducted: (1) using a lower cut-off of ≥3; (2) considering a score of ≥5 on at least one item in the domain (rather than the mean) as reflecting distress. Finally, we explored whether breast cancer survivors had different results compared with women with no history of cancer.

## Patient and public involvement

The authors are thankful to the cancer survivors involved with the Independent Cancer Patients' Voice (http://www.independentcancerpatientsvoice.org.uk/), a patient advocate group, for their comments on the study protocol.

## RESULTS
## Characteristics of the participants

608 women from 40 primary care practices participated in the study (table 1). General practices were from all four UK countries, but there was a predominance of practices from Scotland (N=16) and Wales (N=15) (online supplemental table 2). The median number of consultations in 2018 and 2019 was 11, similar between breast cancer survivors (median 11, IQR: 7–16) and women with no history of cancer (median 11, IQR: 7–20). A quarter of the women had a higher education degree.

## Anxiety and depression

Of the 599 women that replied to the anxiety subscale, 242 (40%) had borderline to abnormal symptoms (table 2). Borderline to abnormal symptoms of depression was also reported by 92 (15%) of the 601 women that replied to the subscale for depression. Almost no women had Read codes for anxiety or depression registered in their EHR in the 24 months prior. However, 108/242 (45%) of those

**Table 1** Characteristics of the study participants*

| | All participants (N=608) | |
|---|---|---|
| | N | % |
| Age at completion of questionnaire | | |
| 34–59 years | 174 | 28.6 |
| 60–69 years | 210 | 34.5 |
| ≥70 years | 224 | 36.8 |
| Highest education level | | |
| Up to GCSEs, O levels or equivalent | 205 | 33.7 |
| A levels or equivalent | 65 | 10.7 |
| Trade or technical training | 106 | 17.4 |
| Undergraduate or postgraduate degree | 160 | 26.3 |
| Did not want to disclose | 72 | 11.8 |
| Ethnicity | | |
| White | 589 | 96.9 |
| Asian/Asian British | 7 | 1.2 |
| Did not want to disclose | 12 | 2.0 |
| IMD quintile | | |
| 1 (least deprived) | 124 | 20.4 |
| 2 | 90 | 14.8 |
| 3 | 81 | 13.3 |
| 4 | 239 | 39.3 |
| 5 (most deprived) | 74 | 12.2 |
| Living arrangements | | |
| Not alone | 458 | 75.3 |
| Alone | 138 | 22.7 |
| Did not want to disclose | 12 | 2.0 |
| Country | | |
| England | 114 | 18.8 |
| Northern Ireland | 49 | 8.1 |
| Scotland | 188 | 30.9 |
| Wales | 257 | 42.3 |

*Proportion may not add to 100% due to rounding.
IMD, index of multiple deprivation.

reporting anxiety symptoms were prescribed an anxiolytic or antidepressant (for anxiety), and 51/92 (55%) of those reporting depression symptoms were prescribed an antidepressant.

In the QoL scale, 100 of the 608 (17%) women that replied to questionnaire had average replies of frequently, very often or always experiencing negative feelings (mean score ≥5). Only one patient (1%) had Read codes related to anxiety and/or depression recorded in their EHR in the 24 months prior, but 51 (51%) had an antidepressant and/or anxiolytic prescription.

Of the patients that had information about negative feelings in their EHR, only a minority reported distress in the questionnaires (online supplemental table 3).

**Table 2** Capture of patient-reported anxiety and depression in patients' primary care records

| Patient-reported outcomes | | Read codes for symptoms/diagnoses in the patients' electronic health records by time prior to LCD | | | | | | | | Relevant drug prescription by time prior to LCD* | | | | | | | |
| | | 3 months | | 6 months | | 12 months | | 24 months | | 3 months | | 6 months | | 12 months | | 24 months | |
| | N | % | N | % | N | % | N | % | N | % | N | % | N | % | N | % | N | % |
| **HADS** | | | | | | | | | | | | | | | | | | |
| Anxiety | | | | | | | | | | | | | | | | | | |
| Normal | 357 | 59.6 | 0 | 0 | 0 | 0 | 0 | 0 | 0 | 0 | 53 | 14.9 | 55 | 15.4 | 67 | 18.8 | 84 | 23.5 |
| Borderline | 124 | 19.7 | 0 | 0 | 0 | 0 | 0 | 0 | 0 | 0 | 33 | 28.0 | 34 | 33.9 | 40 | 33.9 | 49 | 41.5 |
| Abnormal | 118 | 20.7 | 1 | 0.8 | 1 | 0.8 | 2 | 1.6 | 2 | 1.6 | 40 | 32.3 | 42 | 28.8 | 53 | 42.7 | 59 | 50.0 |
| Depression | | | | | | | | | | | | | | | | | | |
| Normal | 509 | 84.7 | 0 | | 0 | | 0 | | 0 | | 75 | 14.7 | 77 | 15.1 | 96 | 18.9 | 120 | 23.6 |
| Borderline | 66 | 4.3 | 0 | | 0 | | 0 | | 0 | | 26 | 39.4 | 26 | 39.4 | 30 | 45.5 | 34 | 51.5 |
| Abnormal | 26 | 11.0 | 0 | | 0 | | 0 | | 0 | | 13 | 50.0 | 13 | 50.0 | 16 | 61.5 | 17 | 65.4 |
| **QLACS** | | | | | | | | | | | | | | | | | | |
| Negative feelings | | | | | | | | | | | | | | | | | | |
| ≥5 | 100 | 16.8 | 1 | 1.0 | 1 | 1.0 | 1 | 1.0 | 1 | 1.0 | 36 | 36.0 | 37 | 37.0 | 47 | 47.0 | 51 | 51.0 |
| ≥3 | 386 | 64.7 | 1 | 0.3 | 1 | 0.3 | 2 | 0.5 | 2 | 0.5 | 89 | 23.3 | 90 | 23.1 | 109 | 28.2 | 129 | 33.4 |
| 1 item ≥5 | 227 | 37.3 | 1 | 0.4 | 1 | 0.4 | 1 | 0.4 | 1 | 0.4 | 65 | 28.6 | 66 | 29.1 | 82 | 36.1 | 95 | 41.9 |

*Anxiolytics or antidepressants for anxiety; antidepressants for depression.
HADS, Hospital Anxiety and Depression Scale; LCD, last collection date for the practice; PRO, patient-reported outcomes; QLACS, Quality of Life in Adults Cancer Survivors Scale.

**Table 3** Capture of patient-reported QoL-related distress in patients' primary care records

| QoL Domain | Domain score | Patient-reported outcomes | | Relevant Read codes in the electronic health record*, by time prior to the date of last data collection | | | | | | | |
| | | | | 3 months | | 6 months | | 12 months | | 24 months | |
| | | No.† | % | No. | % | No. | % | No. | % | No. | % |
|---|---|---|---|---|---|---|---|---|---|---|---|
| Cognitive problems | Average ≥5 | 93 | 15.5 | 0 | | 0 | | 0 | | 0 | |
| | Average ≥3 | 394 | 65.6 | 0 | | 0 | | 0 | | 0 | |
| | 1 item ≥5 | 193 | 31.7 | 0 | | 0 | | 0 | | 0 | |
| Fatigue | Average ≥5 | 157 | 26.1 | 0 | | 0 | | 0 | | 0 | |
| | Average ≥3 | 472 | 78.5 | 0 | | 0 | | 0 | | 0 | |
| | 1 item ≥5 | 536 | 88.2 | 0 | | 0 | | 0 | | 0 | |
| Physical pain | Average ≥5 | 123 | 20.6 | 65 | 52.8 | 70 | 56.9 | 82 | 66.7 | 92 | 74.8 |
| | Average ≥3 | 330 | 55.4 | 106 | 32.1 | 116 | 35.2 | 152 | 46.1 | 186 | 56.4 |
| | 1 item ≥5 | 231 | 38.0 | 86 | 37.2 | 94 | 40.7 | 120 | 52.0 | 142 | 61.5 |
| Sexual dysfunction | Average ≥5 | 156 | 25.7 | 0 | | 0 | | 0 | | 0 | |
| | Average ≥3 | 377 | 62.0 | 0 | | 0 | | 0 | | 0 | |
| | 1 item ≥5 | 304 | 50.0 | 0 | | 0 | | 0 | | 0 | |
| Social avoidance | Average ≥5 | 82 | 13.5 | 0 | | 0 | | 0 | | 0 | |
| | Average ≥3 | 294 | 48.4 | 0 | | 0 | | 0 | | 0 | |
| | 1 item ≥5 | 196 | 32.2 | 0 | | 0 | | 0 | | 0 | |

Severe cognitive dysfunction was an exclusion criterion for the study.
*Relevant Read codes were codes for cognitive impairment, dementia and dementia specific drugs (cognitive problems domain); low energy, tiredness (fatigue domain); pain, pain syndromes, analgesics prescriptions (pain domain); low libido, anorgasm, vaginismus (sexual dysfunction); social isolation and avoidance (social avoidance domain).
†608 women participated in the study; due to missing data for some items, the number of women included in the denominator varies slightly by domain.
QoL, quality of life.

### Other QoL domains: cognitive problems, fatigue, pain, sexual dysfunction, social avoidance

93/608 (16%) women reported high levels of distress related to cognitive problems, 156 (26%) to sexual dysfunction, 157 (26%) to fatigue/energy, and 82 (14%) to social avoidance (table 3). No codes relevant to these domains were found in the patients' EHR up to 24 months prior. Using lower cut-offs to classify patients based on their QoL scores yielded similar results. Distress with pain was reported in the questionnaires by 123 (21%) of the women, and 65 (53%) of these had symptoms of pain and/or analgesic prescription recorded their EHR in the previous 3 months; this increased to 92 (75%) when a longer 24-month time window was used.

### Sensitivity analyses

Results were similar to those of the main analysis when different criteria were used to define distress (see tables 2 and 3). There were no meaningful differences in the results between breast cancer survivors and women with no history of cancer (see online supplemental tables 4 and 5).

### DISCUSSION
### Summary

Most patients who reported clinically relevant symptoms of anxiety and depression, and distress with cognitive problems, fatigue, physical pain, sexual dysfunction and social avoidance, did not have clinical codes for these conditions in their primary care EHR. This suggests that in some cases GPs may be unaware of problems adversely affecting their patients' QoL. Our results may also be partly explained by inconsistent coding, as evident from the number of women in receipt of medications for anxiety and depression, despite no diagnostic codes being present in the EHR. In these cases, GPs were evidently aware of the patient's condition but had not entered a diagnostic code into the EHR, which could lead to misleading information when routine coded data are used for audit and research.

### Strengths and limitations

The ability to compare PROs with data available in the EHR represents a unique strength of this study. To our knowledge, no previous study has reported on this comparison. However, this study had limitations. Data on the date of questionnaire completion or questionnaire return were not available to the research team, and therefore we could not identify, precisely, the consultations that corresponded to when the PROs were evaluated. As questionnaires were returned over a 9-month period, this could have affected our assessment of outcomes particularly in 3–6 months prior. However, even analyses looking

at coding in the previous 24 months showed substantial under-recording of mental-health and QoL-related distress in coded primary care data. Our approach to identify patients experiencing distress on specific QLACS domains used score thresholds that were not validated. However, sensitivity analysis using different cut-offs showed generally the same patterns. CPRD only captures drugs prescribed to patients, and widely used drugs for pain and fatigue are sold over-the-counter. We assumed that anxiolytics and antidepressants were taken for anxiety/depression, but we cannot rule out that some were for other indications such as pain or insomnia. Our definition of fatigue did not include prescriptions, as we did not have information on what drugs were prescribed with the aim of ameliorating this condition. The comparison for cognitive problems was limited by the need to exclude patients unable to reply to a self-reported questionnaire, and we cannot rule out that GPs may have been overly strict in applying this exclusion criterion, excluding mild cognitive impairment. Our results are based on a convenience sample of breast cancer survivors and non-cancer controls and may not be generalisable to the general population; however, results were similar in our sensitivity analysis comparing results between the two groups, which is probably because women with a history of breast cancer were on average 8 years postdiagnosis and most likely not under active treatment for cancer. Half of the patients in our sample had a history of breast cancer, which may have been associated with closer monitoring, and therefore we could have underestimated the extent of the missed coding of these problems. However, four sensitivity analyses comparing those with and without prior cancer showed no major differences between groups. We compared PROs with information coded in the EHR; while most of patient care is coded using records, GPs sometimes use other methods to keep records (eg, free text entry) which were not available to us. Similarly, some codes in the EHR are unspecific (eg, mood observations) and we could not assign a correspondence to domains of QoL. Ford et al[15] explored the reasons for differences in coding for mental health conditions in primary care, and found that GPs may prefer free text and use codes for symptoms or general codes instead of definitive diagnoses. Therefore, it is possible that we underestimated, in some cases, the awareness of the GP about the patients' well-being.

### Comparison with existing literature

Only one in three patients that reported distressing levels of negative feelings had similar information recorded in their EHR in the previous 3 months. This is consistent with patients often not seeking primary care for anxiety and depression.[16] Approximately one-half of the women that reported poor QoL related to pain had related information in the EHR in the previous 3 months. This may be partly explained by patients' self-treating pain with widely used over-the-counter treatments such as paracetamol and ibuprofen. Conversely, the higher recording of pain compared with negative feelings could be related to patients more often seeking help for concerns perceived as being amenable to treatment. Patients with a prescription of antidepressants/anxiolytics and that reported normal levels of depressive/anxiety symptoms are not unexpected—these drugs are effective at improving symptoms of depression and anxiety, but have long treatment courses and patients are recommended to continue pharmacological treatment for months after symptoms disappear to prevent relapse.[17 18]

We did not find any records of cognitive dysfunction, social avoidance, sexual dysfunction or fatigue in the EHR of the participating women in the previous 24 months. An absence of entries for social avoidance is plausible; Read codes for social avoidance have seldom been used in the database. A lack of records for sexual dysfunction is in keeping with evidence that only a small proportion of people contact GPs for issues related to sexual function.[19] The lack of coded records for cognitive dysfunction and fatigue was more unexpected. It is possible that GPs systematically excluded people with mild cognitive dysfunction. For fatigue, a manual review of all entries in the EHR of patients that reported distressing levels of fatigue revealed a pattern of multimorbidity, almost always with diagnoses where fatigue is implicit (eg, heart failure, chronic obstructive pulmonary disease), but no explicit codes for fatigue.

### Implications for research and/or practice

It is important to raise awareness that patients may not always actively report their distress. Even when GPs are aware of health issues, they are not always coded in the patient record, and thus EHRs have low sensitivity to detect patients experiencing poor QoL at a particular point in time. Studies investigating anxiety and depression should consider prescriptions as well as clinical codes, as many patients were prescribed anxiolytics and antidepressants without having a Read code for these conditions.

Similarly to other studies,[20] in this study the collection of PRO data was not followed by feedback of the results to the patients or to the patients' GPs. This was because the authorisation to conduct this study within the UK National Health Service was granted on the basis that there would be separation between the researchers and the identity of patients and GPs, and we could only access anonymised data.[20] Krageloh et al highlight in their review that most studies where there was a formal procedure to feedback PRO results to patients and healthcare providers reported better outcomes in this group compared with controls.[20] Future studies of PRO outcomes in the NHS should explore options to report back results without violating the data protection regulation in place.

### CONCLUSION

We found substantial under-recording of mental-health and QoL-related distress in coded primary care data. In addition, there may be inconsistent coding of known

conditions, meaning that studies of mental-health and QoL-related outcomes using EHR databases likely under-estimate the absolute burden of these outcomes in the population. Further research is needed to understand whether or not GPs are aware of patient distress in cases where codes have not been recorded.

**Contributors** HC, RW and KB designed the study. HC, RW and KB obtained ethical approvals. HD, HC, RW and KB managed data collection. HC entered the data and performed analyses. All authors revised the manuscript for important intellectual content and approved the final version of the manuscript. HC is the guarantor for this study.

**Funding** HC was supported by an MRC Case Studentship in Epidemiology, jointly funded by UK Medical Research Council (MRC; https://mrc.ukri.org/) and the Clinical Practice Research Datalink (https://www.cprd.com/) at the Medicines and Healthcare Products Regulatory Agency (grant number MR/M016234/1 to HC). KB is funded by a Wellcome Senior Research Fellowship (grant number 220283/Z/20/Z) and previously held a Sir Henry Dale Fellowship jointly funded by Wellcome Trust (https://wellcome.ac.uk/) and the Royal Society (https://royalsociety.org/) (grant number 107737/Z/15/Z).

**Competing interests** KB reports grants from Wellcome Trust, the Royal Society, Medical Research Council and British Heart Foundation, outside the submitted work. RW and HD report that CPRD has financial relationships with its clients, including the London School of Hygiene and Tropical Medicine, in relation to providing access to research data and services outside the submitted work.

**Patient and public involvement** Patients and/or the public were involved in the design, or conduct, or reporting, or dissemination plans of this research. Refer to the Methods section for further details.

**Patient consent for publication** Not applicable.

**Provenance and peer review** Not commissioned; externally peer reviewed.

**Data availability statement** Data may be obtained from a third party and are not publicly available. This study is based in part on data from the Clinical Practice Research Datalink obtained under licence from the UK Medicines and Healthcare products Regulatory Agency. The terms of our licence to access the data preclude us from sharing individual patient data with third parties. The electronic health records raw data may be requested directly from CPRD following their usual procedures.

**ORCID iD**
Helena Carreira http://orcid.org/0000-0003-1538-2526

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
