## [Reviewer comments · BMJ Open]

ARTICLE DETAILS

TITLE (PROVISIONAL)	Recording of patients' mental health and quality of life-related outcomes in primary care: a cross-sectional study in the UK
AUTHORS	Carreira, Helena; Williams, Rachael; Dempsey, Harley; Bhaskaran, Krishnan

VERSION 1 – REVIEW

REVIEWER	Hannigan, A University of Limerick
REVIEW RETURNED	17-Aug-2022

GENERAL COMMENTS	This paper addresses the important topic of assessing the strengths and limitations of routinely collected data in primary care for research on mental health and quality of life. There are, however, a number of issues which limit the potential contribution of this paper. The abstract doesn't mention that half the sample are breast cancer survivors - the first mention of this is in the aims of the study. Given this is relevant for the study design and the use of cancer-specific quality of life measures, this information should be given in the abstract. Looking at the already published paper, the breast cancer survivors were on average 8 years post diagnosis and this seems important information to add to this paper. Knowing this makes it less surprising that the sensitivity analysis revealed no major differences between the two groups of women (though there are some e.g. sexual dysfunction). It also makes it more reasonable to assume that the breast cancer survivors were not being provided with significant care by other healthcare professionals outside of general practice which would not be captured in their records. It would be helpful in the introduction to provide references for the statements about coding in general practice (paragraph 2) and in general, the reference list seems short and [13] old. Is there any evidence specific to coding for women attending general practice for mental health issues? This may provide a stronger rationale for exploring this dataset (other than the data are already available). Given the focus of this study, not having the date of completion of the questionnaire is a major limitation particularly given the length of recruitment time from Jan to Nov 2019. Is the date of return of the questionnaire to the researchers available as a proxy? It seems from the description that the two years of EHRs reviewed is 2018 and 2019 (given how little variation there is in date of last data collection from the practice). So that is the same for patients regardless of whether they completed the questionnaire in Jan
--

	2019 or Nov 2019? This would be important to clarify. The authors state that not knowing the date of completion of the questionnaire probably doesn't matter because there aren't distinct differences < 3 to < 24 months but Table 2 shows, for example, rates of prescribing of 32% for 3 months increasing to 50% at 24 months for those with abnormal HADS. We are also not provided with information on attendance patterns for the women in the previous two years. What was the distribution of time from the most recent practice visit for the women to date of last data collection? What was the distribution of the number of attendances by the women over the previous two years? Were there any differences between breast cancer survivors and controls for this? It is also interesting to note that 15% of those with normal HADS and normal depression had a prescription within 3 months. This hasn't been discussed by the authors. Also, given the overlap for fatigue, low libido and social isolation with depression, it may be worth exploring the association with prescriptions in addition to read codes for these QoL domains. Given the sample size of 608 from 40 general practices, numbers are likely too small to observe any differences in coding across the 40 practices but it would be helpful to know more about the characteristics of the practices involved for generalisability. Were there any differences in characteristics between the 40 practices who agreed to participate and the practices that didn't? This conclusion seems very strong given for example the high rate of prescribing "We found substantial under-recording of mental-health and QoL-related distress in coded primary care data, suggesting that there may be missed opportunities to provide support to patients in need." It seems more reasonable to conclude that coding isn't happening rather than care is not being provided and a more nuanced discussion for the reasons why would be helpful e.g. Ford, E., Champion, A., Chamles, D. A., Habash-Bailey, H., & Cooper, M. (2016). "You don't immediately stick a label on them": a qualitative study of influences on general practitioners' recording of anxiety disorders. BMJ open, 6(6), e010746. In general, more data, discussion and acknowledgement of limitations is needed and it may not be reasonable to use this dataset for this purpose given the date of administering the questionnaire is unknown.
--	--

REVIEWER	Turvey, Carolyn Veterans Affairs Health Care System, Iowa City; Department of Psychiatry, Carver College of Medicine, University of Iowa
REVIEW RETURNED	21-Sep-2022

GENERAL COMMENTS	For the most part, this is an excellent paper exploring the correspondence between medical record documentation and indication of mental health impairment in standard questionnaires of patient-reported outcomes. The authors are thorough and the manuscript is well-written. The one overarching comment about the paper is that it does not adequately discuss the repeated finding that PROs are collected
--

	but not always reviewed by their provider. See Krageloh et al. Psychiatric Services 2015. Moreover, collecting this data and not reviewing it with both patient and provider means the PRO collection has little to no impact on outcomes. The specificity of the sample is mentioned in the limitations section and it is a major drawback. It is not just a specific sample, but it is a specific clinical context- one where well being tends to be monitored closely. I suspect this means that the gaps in detection are underestimates and even more levels of distress are not documented than in this study. Otherwise this is an excellent and detailed study. Perhaps, the number of tables are excessive and overly complex, but if there is space fo them- they do make contributions. Carolyn Turvey
--	---

VERSION 1 – AUTHOR RESPONSE

Reviewer #1: Prof. A Hannigan, University of Limerick

Reviewer #1, comment #1

This paper addresses the important topic of assessing the strengths and limitations of routinely collected data in primary care for research on mental health and quality of life. There are, however, a number of issues which limit the potential contribution of this paper.

The abstract doesn't mention that half the sample are breast cancer survivors - the first mention of this is in the aims of the study. Given this is relevant for the study design and the use of cancer-specific quality of life measures, this information should be given in the abstract. Looking at the already published paper, the breast cancer survivors were on average 8 years post diagnosis and this seems important information to add to this paper. Knowing this makes it less surprising that the sensitivity analysis revealed no major differences between the two groups of women (though there are some e.g. sexual dysfunction). It also makes it more reasonable to assume that the breast cancer survivors were not being provided with significant care by other healthcare professionals outside of general practice which would not be captured in their records.

Authors' reply:

Thank you for reviewing our paper. We agree that it's important to clarify in the abstract that half of the sample included women with history of breast cancer. The revised abstract is provided on page 2, and a transcription is provided below:

"Participants: 608 women registered in the Clinical Practice Research Datalink (CPRD) GOLD primary care database (a convenience sample using data from a previous study on 356 breast cancer survivors (8.1 years post diagnosis) and 252 women with no prior cancer)."

We also added to the discussion the point about breast cancer survivors being most likely not under active treatment for cancer. Please see page 15; for convenience, the revised text is also provided below:

"Our results are based on a convenience sample of breast cancer survivors and non-cancer controls and may not be generalisable to the general population; however, results were similar in our sensitivity analysis comparing results between the two groups, which is probably because women with history of breast cancer were on average 8 years post diagnosis and most likely not under active treatment for cancer."

Reviewer #1, comment #2

It would be helpful in the introduction to provide references for the statements about coding in general practice (paragraph 2) and in general, the reference list seems short and [13] old. Is there any evidence specific to coding for women attending general practice for mental health issues? This may provide a stronger rationale for exploring this dataset (other than the data are already available).

Authors' reply:

Thank you. The issue of underdiagnosis and undertreatment of common mental health disorders in primary care is highlighted in the UK National Institute for Health and Care Excellence (NICE) Clinical Guideline for "Common mental health problems: identification and pathways to care". We have added this reference to the introduction (page 6). An excerpt of the text is provided below:

"There has been a lack of research quantifying the burden of mental health and other QoL-related complaints that have not been picked up in primary care, and therefore may remain undiagnosed and untreated [6]."

6. National Institute for Health and Care Excellence (NICE), *Common mental health problems: identification and pathways to care. Clinical guideline 123*. 2011.

In addition, we added two further references in the introduction to support our sentences:

Under-recording of problems in primary care records could suggest lack of awareness by the general practitioner (GP) about the patient's lack of wellbeing, and thus a missed opportunity for care. Under-recording might also reflect inconsistent coding of mental health and QoL problems in the primary care record, with important implications for audit and research based on electronic health records (EHR) [7, 8].

7. Gray, J., D. Orr, and A. Majeed, *Use of Read codes in diabetes management in a south London primary care group: implications for establishing disease registers*. *BMJ*, 2003. 326(7399): p. 1130.

8. Shephard, E., S. Stapley, and W. Hamilton, *The use of electronic databases in primary care research*. *Fam Pract*, 2011. 28(4): p. 352-4.

Lastly, we replaced former reference 13 (now ref 19) with a more recent one (see below).

19. Hinchliff, S., et al., Seeking help for sexual difficulties: findings from a study with older adults in four European countries. *Eur J Ageing*, 2020. 17(2): p. 185-195.

Reviewer #1, comment #3:

Given the focus of this study, not having the date of completion of the questionnaire is a major limitation particularly given the length of recruitment time from Jan to Nov 2019. Is the date of return of the questionnaire to the researchers available as a proxy? It seems from the description that the two years of EHRs reviewed is 2018 and 2019 (given how little variation there is in date of last data collection from the practice). So that is the same for patients regardless of whether they completed the questionnaire in Jan 2019 or Nov 2019? This would be important to clarify. The authors state that not knowing the date of completion of the questionnaire probably doesn't matter because there aren't distinct differences < 3 to < 24 months but Table 2 shows, for example, rates of prescribing of 32% for 3 months increasing to 50% at 24 months for those with abnormal HADS.

Authors' reply:

Unfortunately the date of return of the questionnaires is not available either; this is because the authors were not able to receive the questionnaires directly (in case of accidental disclosure of participant identities), and dates of return were not available from the CPRD team who processed the questionnaires. Without a date of questionnaire completion, or a date of questionnaire return to CPRD, we could not differentiate between those that completed the questionnaires in January versus in November. We agree this is an important limitation of our study and we have expanded the discussion to be transparent. Please see page 15, where we added:

"We did not collect data on the date of questionnaire completion or questionnaire return to the research team, and therefore we could not identify, precisely, the consultations that corresponded to when the PROs were evaluated. As questionnaires were returned over a 9-month period, this could

have affected our assessment of outcomes particularly in 3 to 6 months prior. However, even analyses looking at coding in the previous 24 months showed substantial under-recording of mental-health and QoL-related distress in coded primary care data.”

Reviewer #1, comment #4:

We are also not provided with information on attendance patterns for the women in the previous two years. What was the distribution of time from the most recent practice visit for the women to date of last data collection? What was the distribution of the number of attendances by the women over the previous two years? Were there any differences between breast cancer survivors and controls for this?

Authors' reply:

Thank you for this comment. There were no important differences in the median number of consultations in 2018 and 2019 for women in the breast cancer survivor and control groups. We have added these results to the manuscript. Please see page 10 and transcription below:

“The median number of consultations in 2018 and 2019 was 11, similar between breast cancer survivors (median 11, inter-quartile range (IQR): 7-16) and women with no history of cancer (median 11, inter-quartile range (IQR): 7-20).”

Reviewer #1, comment #5:

It is also interesting to note that 15% of those with normal HADS and normal depression had a prescription within 3 months. This hasn't been discussed by the authors. Also, given the overlap for fatigue, low libido and social isolation with depression, it may be worth exploring the association with prescriptions in addition to read codes for these QoL domains.

Authors' reply:

Thank you for pointing this out. We have now added the following text to the discussion, on p. 16:

“Patients with a prescription of antidepressants / anxiolytics and that reported normal levels of depressive / anxiety symptoms are not unexpected – these drugs are effective at improving symptoms of depression and anxiety, but have long treatment courses and patients are recommended to continue pharmacological treatment for months after symptoms disappear to prevent relapse [16, 17].”

We agree it would be interesting to look at the association between prescriptions and fatigue, low libido and social isolation. Unfortunately, this was not covered in our study protocol, and it is unlikely that we would obtain permission to use the data for this purpose because the small numbers of events are likely insufficient to do analysis with reasonable statistical power to detect an association, should it exist.

Reviewer #1, comment #6

Given the sample size of 608 from 40 general practices, numbers are likely too small to observe any differences in coding across the 40 practices but it would be helpful to know more about the characteristics of the practices involved for generalisability. Were there any differences in characteristics between the 40 practices who agreed to participate and the practices that didn't?

Authors' reply:

We added a table (new Supplementary table 2) with the characteristics of the 40 practices that participated in the study. We also compared this information with the overall distribution of practices in the CPRD entire database. The following sentence was added to the results, on p. 10:

“General practices were from all four UK countries, but there was a predominance of practices from Scotland (N=16) and Wales (N=15) (Supplementary Table 2).”

Reviewer #1, comment #7

This conclusion seems very strong given for example the high rate of prescribing “We found substantial under-recording of mental-health and QoL-related distress in coded primary care data, suggesting that there may be missed opportunities to provide support to patients in need.” It seems more reasonable to conclude that coding isn’t happening rather than care is not being provided and a more nuanced discussion for the reasons why would be helpful e.g.

Ford, E., Campion, A., Chamles, D. A., Habash-Bailey, H., & Cooper, M. (2016). “You don’t immediately stick a label on them”: a qualitative study of influences on general practitioners’ recording of anxiety disorders. *BMJ open*, 6(6), e010746.

Authors’ reply:

Thank you. We changed our conclusion to:

“We found substantial under-recording of mental-health and QoL-related distress in coded primary care data. In addition, there may be inconsistent coding of known conditions, meaning that studies of mental-health and QoL-related outcomes using EHR databases likely underestimate the absolute burden of these outcomes in the population. Further research is needed to understand whether or not GPs are aware of patient distress in cases where codes have not been recorded.”

We also changes the abstract conclusion to:

“Conclusion: Many patients reporting mental health and QoL problems had no record of this in coded primary care data. This suggests that coded data does not fully reflect the burden of disease. Further research is needed to understand whether or not GPs are aware of patient distress in cases where codes have not been recorded.”

In addition, we added the following sentence to the discussion, p. 16:

“Ford et al. [15] explored the reasons for differences in coding for mental health conditions in primary care, and found that GPs may prefer free text and use codes for symptoms or general codes instead of definitive diagnoses. Therefore, it is possible that we underestimated, in some cases, the awareness of the GP about the patients’ wellbeing.”

Reviewer #1, comment #8:

In general, more data, discussion and acknowledgement of limitations is needed and it may not be reasonable to use this dataset for this purpose given the date of administering the questionnaire is unknown.

Authors’ reply:

We acknowledge that our study has limitations; we have been transparent in disclosing these and we have expanded our discussion of the potential implications in this revision. However, we believe our study is unique and valuable in being able to compare data collected directly from patients with information recorded in their coded clinical record; to our knowledge this has not been done in any comparable study. As noted in our response to comment #3 above, even when we look at a relatively long lookback period for codes (which should be less affected by not knowing the exact questionnaire date), our conclusion that coded EHR data substantially underestimates the burden of mental health conditions is still supported. Considering the very high burden of mental health conditions in the population, knowledge on this is an important contribution for public health.

Reviewer #2: Dr. Carolyn Turvey, Veterans Affairs Health Care System, Iowa City; Department of Psychiatry, Carver College of Medicine, University of Iowa

Reviewer #2, comment #1:

For the most part, this is an excellent paper exploring the correspondence between medical record documentation and indication of mental health impairment in standard questionnaires of patient-reported outcomes. The authors are thorough and the manuscript is well-written.

Authors’ reply:

Thank you.

Reviewer #2, comment #2:

The one overarching comment about the paper is that it does not adequately discuss the repeated finding that PROs are collected but not always reviewed by their provider. See Krageloh et al. Psychiatric Services 2015. Moreover, collecting this data and not reviewing it with both patient and provider means the PRO collection has little to no impact on outcomes.

Authors' reply:

Thank you for bringing this point to our attention. We now discuss it in the manuscript, p.17; see transcript below:

“Similarly to other studies [20], in this study the collection of PRO data was not, unfortunately, followed by feedback of the results to the patients or to the patients' GPs. This was because the authorisation to conduct this study within the UK National Health Service was granted on the basis that there would separation between the researchers and the identity of patients and GPs, and we could only access anonymised data. Krageloh et al., 2015 [20] highlight in their review that most studies where there was a formal procedure to feedback PRO results to patients and health care providers reported better outcomes in this group compared to controls [20]. Future studies of PRO outcomes in the NHS should explore options to report back results without violating the data protection regulation in place.”

Reviewer #2, comment #3:

The specificity of the sample is mentioned in the limitations section and it is a major drawback. It is not just a specific sample, but it is a specific clinical context- one where well being tends to be monitored closely. I suspect this means that the gaps in detection are underestimates and even more levels of distress are not documented than in this study.

Authors' reply:

Thank you. We agree and added this point to the discussion, p. 16:

“Half of the patients in our sample had history of breast cancer, which may have been associated with closer monitoring, and therefore we could have underestimated the extent of missed coding of these problems. However, four sensitivity analysis comparing those with and without prior cancer showed no major differences between the groups.”

Reviewer #2, comment #4:

Otherwise this is an excellent and detailed study. Perhaps, the number of tables are excessive and overly complex, but if there is space for them- they do make contributions.

Authors' reply:

Thank you.

VERSION 2 – REVIEW

REVIEWER	Hannigan, A University of Limerick
REVIEW RETURNED	04-Nov-2022

GENERAL COMMENTS	Most of my comments have been adequately addressed. There are a number of limitations to this study but these have now been listed in the paper. It may be worth specifying in the list of strengths and limitations that this is a convenience sample i.e. 'This study included only a convenience sample of adult women and the results may not be generalizable to all women, men or to other age groups.'
---

	A study protocol was mentioned but I don't see this included as a supplementary file for this paper or a reference to it? It wasn't mentioned in the previous version. The only characteristic compared for the general practices who participated was geographical location - is this the only one available? There is no information on practice size for example? It seems a limitation not to have included a request for prescriptions with read codes for fatigue in particular - this could also be added to limitations. A power calculation is mentioned in the author's response but I couldn't see its relevance here - was this for the original study which explored differences between women with breast cancer and controls?
REVIEWER	Turvey, Carolyn Veterans Affairs Health Care System, Iowa City; Department of Psychiatry, Carver College of Medicine, University of Iowa
REVIEW RETURNED	28-Oct-2022
GENERAL COMMENTS	This is an attentive revision where the authors acknowledge the role of underdocumentation even when the provider recognizes a problem. They also acknowledge that it is a very select sample and a select clinical use case.

VERSION 2 – AUTHOR RESPONSE

Reviewer #1: Prof. A Hannigan, University of Limerick

Reviewer #1, comment #1

“Most of my comments have been adequately addressed.

There are a number of limitations to this study but these have now been listed in the paper. It may be worth specifying in the list of strengths and limitations that this is a convenience sample i.e. 'This study included only a convenience sample of adult women and the results may not be generalizable to all women, men or to other age groups.’”

Authors' reply:

Thank you for reviewing our paper. We have updated the strengths and limitations as suggested. The bullet point now reads:

- *This study included only a convenience sample of adult women and the results may not be generalizable to all women, men or to other age groups.*

Reviewer #1, comment #2

“A study protocol was mentioned but I don't see this included as a supplementary file for this paper or a reference to it? It wasn't mentioned in the previous version.”

Authors' reply:

We added the information on the study protocol to address an editorial comment that asked us to complete the STROBE checklist, and this includes an item to mention how we arrived at the sample size. The study protocol had been provided as supplementary information for the editor only. We have now included the protocol as a supplementary document to the paper. We reference the protocol in page 6: “The study protocol (Supplementary Materials) provides the sample size calculations for the original study.” Thank you for this comment.

Reviewer #1, comment #3

“The only characteristic compared for the general practices who participated was geographical location - is this the only one available? There is no information on practice size for example?”

Authors' reply:

Thank you. We included practice size in supplementary table 2. We don't have more information about the practices that did not participate.

Reviewer #1, comment #4

“It seems a limitation not to have included a request for prescriptions with read codes for fatigue in particular - this could also be added to limitations. A power calculation is mentioned in the author's response but I couldn't see its relevance here - was this for the original study which explored differences between women with breast cancer and controls?”

Authors' reply:

Thank you. We added the following sentence to the limitations: “Our definition of fatigue did not include prescriptions, as we did not have information on what drugs were prescribed with the aim of ameliorating this condition.” The power calculations mentioned were for the original study – we have clarified this in the text, which now reads: “The study protocol (Supplementary Materials) provides the sample size calculations for the original study.”

Reviewer #2: Dr. Carolyn Turvey, Veterans Affairs Health Care System, University of Iowa

Reviewer #2, comment #1:

“This is an attentive revision where the authors acknowledge the role of underdocumentation even when the provider recognizes a problem. They also acknowledge that it is a very select sample and a select clinical use case.”

Authors' reply:

Thank you for reviewing our paper.

VERSION 3 – REVIEW

REVIEWER	Hannigan, A University of Limerick
REVIEW RETURNED	29-Nov-2022

GENERAL COMMENTS	All of the comments have been addressed.
--